# Astrocytes Modulate Somatostatin Interneuron Signaling in the Visual Cortex

**DOI:** 10.3390/cells11091400

**Published:** 2022-04-20

**Authors:** Vanessa Jorge Henriques, Angela Chiavegato, Giorgio Carmignoto, Marta Gómez-Gonzalo

**Affiliations:** 1Neuroscience Institute, National Research Council (CNR-IN), 35131 Padua, Italy; vanessajhenriques@gmail.com (V.J.H.); gcarmi@bio.unipd.it (G.C.); 2Department of Biomedical Science, University of Padua, 35131 Padua, Italy; angela.chiavegato@unipd.it

**Keywords:** astrocyte, gliotransmission, somatostatin, calcium, disinhibition

## Abstract

At glutamatergic synapses, astrocytes respond to the neurotransmitter glutamate with intracellular Ca^2+^ elevations and the release of gliotransmitters that modulate synaptic transmission. While the functional interactions between neurons and astrocytes have been intensively studied at glutamatergic synapses, the role of astrocytes at GABAergic synapses has been less investigated. In the present study, we combine optogenetics with 2-photon Ca^2+^ imaging experiments and patch-clamp recording techniques to investigate the signaling between Somatostatin (SST)-releasing GABAergic interneurons and astrocytes in brain slice preparations from the visual cortex (VCx). We found that an intense stimulation of SST interneurons evokes Ca^2+^ elevations in astrocytes that fundamentally depend on GABA_B_ receptor (GABA_B_R) activation, and that this astrocyte response is modulated by the neuropeptide somatostatin. After episodes of SST interneuron hyperactivity, we also observed a long-lasting reduction of the inhibitory postsynaptic current (IPSC) amplitude onto pyramidal neurons (PNs). This reduction of inhibitory tone (i.e., disinhibition) is counterbalanced by the activation of astrocytes that upregulate SST interneuron-evoked IPSC amplitude by releasing ATP that, after conversion to adenosine, activates A_1_Rs. Our results describe a hitherto unidentified modulatory mechanism of inhibitory transmission to VCx layer II/III PNs that involves the functional recruitment of astrocytes by SST interneuron signaling.

## 1. Introduction

The glial cell astrocytes exert key roles in brain functions, from the regulation of synaptic transmission to the control of neurovascular coupling and behavioral responses [1,2,3,4]. Over the last three decades, accumulating evidence shows that astrocytes express a large variety of membrane receptors that sense extracellular signals, including neurotransmitters such as glutamate, adenosine, norepinephrine, γ-aminobutyric acid (GABA), histamine, adenosine triphosphate (ATP), and acetylcholine [5,6,7,8]. Activation of astrocytic metabotropic or ionotropic receptors by neuronal signals results in complex changes in cytosolic Ca^2+^ levels in astrocytes that ultimately trigger the release of various gliotransmitters, contributing to synaptic transmission modulation [6,9]. Although the role of astrocytes at GABAergic synapses has not been thoroughly investigated, a growing body of evidence is highlighting the importance and the circuit-specific complexity of astrocyte-interneuron communication [10,11,12,13,14,15,16,17,18,19,20].

It is known that cytosolic Ca^2+^ elevation at the soma is an indicator of astrocytic responsiveness to episodes of intense neuronal activity, leading to the idea that astrocytes can detect information deriving from intense neuronal activity, while failing to detect low levels of synaptic activity [6]. However, recent studies that used genetically encoded Ca^2+^ indicators (GECIs) revealed an impressive complexity of changes in Ca^2+^ dynamics, spatially localized at fine astrocytic processes, including perisynaptic astrocytic processes (PAPs), the so-called “microdomains” [21]. Microdomains may exhibit Ca^2+^ elevations as fast as elevations observed in neurons [22] and they have been proposed to integrate the activity of several individual synapses. However, the mechanisms that regulate Ca^2+^ dynamics in microdomains and their relationship to receptor-mediated Ca^2+^ responses to neurotransmitters remain poorly defined.

In the cerebral cortex, neuronal networks are mainly composed of glutamatergic excitatory neurons and GABAergic inhibitory interneurons [23]. The diversity of GABAergic interneurons, in terms of morphology, connectivity, and molecular and functional properties, ensures a specific signaling to nearby neurons, controlling local network excitability and modulating synaptic transmission [24,25]. A major subclass of GABAergic interneurons is the somatostatin (SST)-expressing interneuron class that controls signal integration and synaptic plasticity by targeting the distal dendrites of pyramidal neurons (PNs) [26].

We recently described a signaling specificity between different interneurons and astrocytes in the somatosensory cortex (SSCx), where astrocytes respond differently with Ca^2+^ elevations to SST- and parvalbumin (PV)-expressing interneuron activity and exhibit a remarkable plasticity only in response to SST interneuron signaling [27]. The different interneuron classes can, therefore, activate astrocytes differently, raising questions about the specific functional role of this specific SST interneuron-to-astrocyte signaling.

Here, we investigate whether in the visual cortex (VCx), where SST interneurons play a central role in the regulation of synaptic circuits [26,28], the release of GABA from SST interneurons induces a Ca^2+^ response in astrocytes. We also evaluate whether this activation of astrocytes results in a modulation of the inhibitory signal of SST interneurons to PNs. To address these issues, we used patch-clamp recordings in VCx slices and an adeno-associated virus (AAV)-based strategy that combines optogenetics with 2-photon Ca^2+^ imaging experiments. We describe a complex mechanism of reciprocal functional interactions between astrocytes and SST interneurons that modulates the inhibitory tone in VCx circuits.

Our findings provide evidence that VCx astrocytes are recruited by SST interneurons and modulate the inhibitory signaling of these interneurons onto PNs. The identification of the functional significance of interneuron-to-astrocyte signaling opens new perspectives in our understanding of the cellular mechanism that controls the inhibitory tone in brain circuits.

## 2. Materials and Methods

### 2.1. Animal Strains

The mice used in this study were transgenic mice Sst < tm2.1(cre)Zjh > (SST-Cre) and IP_3_R2KO::SST-Cre line (obtained by crossing (IP_3_R2KO mice [29] and SST-Cre mice). All procedures were conducted in accordance with the Italian and European Communities Council Directive of Animal Care and were approved by the Italian Ministry of Health.

### 2.2. Virus Injection

Injections of adeno-associated viruses (AAVs, 250 nL per injection) were performed into the VCx of newborn mice (P1–P2) that had been anaesthetized on ice and secured into a molded platform. Using a manually graduated pulled glass pipette, connected to a custom pressure injection system, we punched the skull and unilaterally injected the viral vector. The stereotaxic coordinates for VCx were 0.5 mm posterior to Lambda, 1.5 mm lateral to sagittal sinus and 0.3 mm in depth. After the injections, the skin was sutured and the pups were revitalized under a heat lamp and returned to their cage. AAVs AAV2/1.EF1.dflox.hChR2(H134R)-mCherry.WPRE.hGH (6.06 × 10^12^ vg/mL, Addgene, Watertown, MA, USA), carrying the doublefloxed light-gated cation channel Channelrhodopsin-2 (ChR2) sequence, and AAV5.GfaABC1DcytoGCaMP6f.SV40 (1.81 × 10^13^ vg/mL, Addgene), carrying the astrocytic promoter GfaABC1D, which induces a sparse expression of the Ca^2+^ indicator GCaMP6f in astrocytes, were injected at a 3:2 ratio. 

### 2.3. Brain Slice Preparation

Coronal brain slices (350 µm) containing the VCx were obtained two weeks after viral injection. The animals were anesthetized with isoflurane and the brains were removed and transferred into an ice-cold artificial cerebrospinal fluid (ACSF) containing (in mM): 125 NaCl, 2.5 KCl, 2 CaCl_2_, 1 MgCl_2_, 25 glucose, 25 NaHCO_3_, 1.25 NaH_2_PO_4_, pH 7.4 with 95% O_2_ and 5% CO_2_). Slices were cut with a vibratome (Leica Vibratome VT1000S, Wetzlar, Germany) in the ice-cold solution described in Dugue et al. 2005 [30], containing (in mM) 130 KGluconate, 15 KCl, 0.2 EGTA, 20 HEPES, 25 glucose, and 2 kynurenic acid. This solution, which mimics the intracellular medium, was used to limit the entry of calcium and other extracellular ions into neurons damaged during the slicing procedure. The kynurenic acid was added to prevent glutamate excitotoxicity. The slices were then transferred for 1 min in a room-temperature solution containing (in mM): 225 D-mannitol, 2.5 KCl, 1.25 NaH_2_PO_4_, 26 NaHCO_3_, 25 glucose, 0.8 CaCl_2_, and 8 MgCl_2_ with 95% O_2_ and 5% CO_2_. Finally, the slices were transferred in ACSF at 30 °C for 15 min and then maintained at room temperature for the entire experiment. In a set of experiments, to confirm the expression of GCaMP6f in astrocytes, slices were kept in ACSF with sulforhodamine 101 (SR101) (0.1 μM, Sigma Aldrich, Milano, Italy) at 32 °C for 15 min to selectively stain astrocytes. 

### 2.4. Ca^2+^ Imaging and Optogenetic Stimulation

To image the Ca^2+^ dynamics in GCaMP6f-expressing astrocytes in brain slices, we used a 2-photon laser scanning microscope Multiphoton Imaging System (Scientifica Ltd., Uckfield, East Sussex, UK) equipped with a pulsed infrared laser (Chameleon Ultra 2, Coherent, Santa Clara, CA, USA). Power at sample was controlled in the range of 5–15 mW to avoid photostimulation and photobleaching. The excitation wavelengths used were 920 nm for GCaMP6f and 780 nm for mCherry. Images were acquired with a water-immersion lens (Olympus, LUMPlan FI/IR ×20, 1.05 numerical aperture (NA)) at a resolution of 512 × 512 pixels, with zoom 2 or 3 (150 × 150 or 120 × 120 µm). We performed recordings of 2 min at a 1.53 Hz acquisition frame rate, with time intervals of 3 min between recordings. Imaging was performed in layers II/III of the VCx. We selected these layers because Martinotti cells, which represent the larger fraction of all SST interneurons, are found across cortical layers II-VI. Even though they are most abundant in layer V, they present an ascending axonal projection that projects vertically, mostly toward layer I with some toward layer II/III, contacting with the distal dendrites of PNs [31]. For optogenetic stimulation, we used full-field photo-stimulation of ChR2-expressing SST interneurons, consisting of trains of 30 light pulses at 1 Hz (λ = 470 nm, 150 ms pulse duration, 3–4 mW at sample plane), blue LED-COM1 (Scientifica Ltd., Uckfield, East Sussex, UK), delivered to the sample through the objective. The 30 light pulse stimulation protocol was repeated twice. This type of optogenetic stimulation protocol induced in ChR2-expressing SSCx SST interneurons in vivo a firing activity comparable to that exhibited by these interneurons in awake mice [27].

#### Ca^2+^ Analysis

Image sequences were aligned and processed with ImageJ and MATLAB software. Detection of astrocyte regions of interest (ROIs) containing Ca^2+^ elevations was performed with ImageJ in a semi-automated manner using the GECIquant plugin [32]. The software was used to identify ROIs corresponding first to the soma (>30 μm^2^; confirmed by visual inspection), then to the proximal processes (>20 μm^2^ excluding soma, corresponding to thicker branches departing from the somatic region), and finally to the microdomains (between 2 and 20 μm^2^, corresponding neither to the soma nor the proximal processes). All pixels within each ROI were averaged to give a single time course F(t). Analyses of Ca^2+^ signals were performed with ImageJ (NIH) and a modified version of a custom software developed in MATLAB 7.6.0 R2008 A (Mathworks, Natick, MA, USA) [27]. The Ca^2+^ signal for each ROI was expressed as ΔF/F_0_ = (F(t) − F_0_)/(F_0_), where F_0_ is defined as the 15th percentile of the whole fluorescent trace for each ROI and considered as a global baseline. For each ROI, we then defined as the baseline trace the points of the ∆F/F_0_ trace with absolute values smaller than twice the standard deviation of the overall signal. Significant Ca^2+^ events were then selected with a supervised algorithm as follows. First, a new standard deviation was calculated on the baseline trace, and all local maxima with absolute values exceeding twice this new standard deviation were identified. Second, of these peaks, we considered as possible significant events only those exceeding a new threshold calculated as four times the standard deviation of the local baseline, which corresponded to the trace between the peak of interest and the previous significant one. The amplitude of each Ca^2+^ event was measured from the 20th percentile of the fluorescent trace interposed between its maximum and the previous significant one.

Basically, this procedure combines a threshold measured from the global baseline with a stricter threshold computed from a local baseline. We adopted this method to reduce artefacts from the recording noise superimposed on the slow astrocytic Ca^2+^ dynamics and from slow changes in the baseline due to physiological or imaging drifts. All Ca^2+^ traces were visually inspected to exclude the ROIs dominated by noise. To quantify the overall microdomain and proximal process activity per astrocyte, the mean amplitude of Ca^2+^ events, the number of active ROIs, and the frequency of Ca^2+^ events per cell were measured under different experimental conditions (basal (bsl), stimulation 1 (S1), recovery 1 (R1), stimulation 2 (S2), and recovery 2 (R2)). Basal values were obtained by calculating the mean values for the different parameters from two basal recordings. To identify the conditions (S1, R1, S2, and R2) that differed from the basal condition (bsl), depending on the normality of the data, we used one-way repeated measures ANOVA with a post hoc Holm-Sidak test or one-way repeated measures ANOVA on ranks with Dunnett’s test.

### 2.5. Electrophysiological Recordings

Brain slices were continuously perfused in a submerged chamber at a rate of 3–4 mL/min with (in mM) NaCl, 120; KCl, 2.5; NaH_2_PO_4_, 1; NaHCO_3_, 26; MgCl_2_, 1; CaCl_2_, 2; and glucose, 10, at pH 7.4 (with 5% CO_2_ and 95% O_2_). Neurons were visualized under a microscope (TCS-SP5-RS, Leica Microsystems, GmbH, Wetzlar, Germany or Olympus FV1000 microscope (Olympus Optical, Tokyo, Japan)) equipped with a CCD camera for differential interference contrast (DIC) image acquisition. Single cell recordings were performed in voltage-clamp configuration using a multiclamp-700B amplifier (Molecular Devices, Foster City, CA, USA). Signals were filtered at 1 kHz and sampled at 10 kHz with a Digidata 1440A interface and pClamp10 software (Molecular Devices, Foster City, CA, USA). Typical pipette resistance was 3–4 MΩ. Whole-cell intracellular pipette solution was (in mM) K-gluconate, 135; KCl, 10; Hepes, 10; MgCl_2_, 1; and Na_2_ATP, 2 (pH 7.4 adjusted with KOH, 280–290 mOsm/L). Series and input resistance were monitored throughout the experiment using a 5 mV pulse. Recordings were considered stable when the change of series and input resistances were below 30%. Cells that did not meet these criteria were discarded. Pyramidal neurons were identified on the basis of their distinct morphology, characterized by a triangular shape of the soma, a main apical dendrite pointing toward the pia, and the absence of a main dendrite in the opposite direction. Their biophysical identity was confirmed by their response to hyperpolarizing and depolarizing 750 ms current steps. In particular, regular spiking pyramidal neurons showed a firing discharge with no spike amplitude accommodation (except for the second action potential in some cells), small after hyperpolarization and low steady-state frequency (15–23 Hz) with 200 pA current injection. ChR2-positive SST interneurons were identified by fluorescence and their biophysical identity was confirmed by a clear sag, one or more rebound action potentials, spike amplitude accommodation and frequency adaptation, and large biphasic after hyperpolarization. The following parameters were measured to characterize passive membrane properties: resting membrane potential (V_rest_) was recorded immediately after the rupture of the neuronal membrane; input resistance (R_in_) was determined by measuring the voltage change in response to small hyperpolarizing and depolarizing current pulses (±50 pA, 750 ms) at resting potential. Evoked inhibitory postsynaptic currents (IPSCs) were recorded in pyramidal cells from cortical layers II/III while holding the membrane potential at −40 mV. IPSCs were evoked by full-field photo-stimulation of ChR2-expressing SST interneurons with brief optogenetic light pulses applied at 0.1 Hz (λ = 470 nm, 4 ms) by a 5-W blue LED (Thorlabs LED1B), which was collimated and coupled under the objective with an optic fiber (ThorLabs, Newton, NJ, USA) held at an angle of 26 degrees. The optic fiber was 300 µm in diameter with a 0.22 NA. The led was directly controlled by a command voltage with a TTL signal. The stimulus intensity was adjusted to evoke 50% maximal IPSC amplitude. The IPSC amplitude was measured as the peak current amplitude minus the mean baseline current before stimulus. Mean IPSCs were grouped in 2.5-min bins (i.e., mean IPSCs from 15 stimuli) to illustrate the time course of IPSC amplitude. After 10 min of basal recording, a train of 30 light pulses (λ = 473 nm, 150 ms pulse duration) was delivered at 1 Hz and maximum intensity (3–4 mWs) to evoke an intense episode of SST interneuron activity. Then, recording of evoked IPSC at 0.1 Hz was resumed. The 30 light pulse stimulation protocol was repeated twice. For statistical analysis, the mean values obtained at basal conditions (10 min recording) were compared to values obtained after SST interneuron stimulation (10 min recording), under the different experimental conditions. When indicated, we performed statistical analysis of IPSC amplitude after SST interneuron stimulation in the absence or presence of antagonists.

### 2.6. Immunohistochemistry and Cell Counting

For the evaluation of the number of ChR2-mCherry-expressing SST interneurons and GCaMP6f-expressing astrocytes, we prepared 70 μm thick slices from brains postfixed overnight at 4 °C in 4% PFA of young (P14–17) SST-Cre mice injected at P0-P2 with AAV2/1.EF1.dflox.hChR2(H134R)-mCherry.WPRE.hGH or AAV5.GfaABC1DcytoGCaMP6f.SV40. Floating sections were incubated for 1 h in the blocking serum (BS: 1% BSA, 2% goat serum, and 1% horse serum in PBS) and 0.2% TritonX-100. For the evaluation of the number of ChR2-mCherry-expressing SST interneurons, slices were incubated with the primary antibodies anti-somatostatin (RRID: AB_2255365, 1:200 rat monoclonal, Merck Millipore, Darmstadt, Germany, MAB354) plus anti-RFP (RRID: AB_2209751, 1:800 rabbit polyclonal, Rockland, Limerick, PA, USA, 600-401-379) for 48 h at 4 °C diluted in the BS, 0.3% TritonX-100 and 0.01% sodium azide. Then, slices were incubated for 2 h at room temperature with secondary antibodies Alexa Fluor 488 conjugated goat anti-rat immunoglobulins pre-adsorbed (RRID: AB_2650997, 1:200, Abcam, Cambridge, UK, ab150165) and goat Alexa Fluor 555 conjugated anti-rabbit F(ab’)2 fragment cross-adsorbed (RRID: AB_2535851, 1:500, Thermofisher Scientific Invitrogen, Waltham, MA, USA, A21430), diluted in BS plus 0.3% TritonX-100. For further controlling specificity of ChR2-mCherry- SST-expressing interneurons, we performed double immunofluorescence with anti-Parvalbumin antibody (RRID:AB_298032, 1:300, rabbit polyclonal, Abcam, ab11427) and monoclonal anti-RFP (RRID:AB_2611063, 1:1000, mouse, Rockland, 200-301-379) with donkey Alexa Fluor 488 conjugated anti-rabbit (RRID:AB_2535792, 1:500, Thermofisher Scientific Invitrogen, A21206) and donkey Alexa Fluor 556 anti-mouse (RRID:AB_2534012, 1:500, Thermofisher Scientific Invitrogen, A10036) cross-adsorbed immunoglobulins. For the evaluation of the number of GCaMP6f-expressing astrocytes, we performed the incubation with primary antibodies mixed and diluted in BS and 0.02% TritonX-100 (16 h at 4 °C). The primary antibodies used were anti-S100β (RRID: AB_2620025, 1:400 guinea pig polyclonal, Synaptic Systems, Goettingen, Germany, 287004) as astrocyte marker and anti-NeuN (RRID: AB_2298772, 1:100, mouse monoclonal, Merck Millipore, MAB377) as neuronal marker. After washing with PBS, slices were incubated for 2 h at room temperature with the secondary antibodies, anti-guinea pig Alexa Fluor 546 (RRID: AB_2534118, 1:500, Thermofisher Scientific Invitrogen, A11074) or anti-mouse Alexa Fluor 556 (RRID: AB_2534012, 1:500, Thermofisher Scientific Invitrogen, A10036), respectively. To enhance the GCaMP6f fluorescence, slices were further incubated overnight at 4 °C with rabbit anti-GFP antibody directly conjugated with Alexa Fluor 488 (RRID: AB_221477, 1:200 rabbit, Thermofisher Scientific Invitrogen, A21311). Slices were then washed 3 times for 10 min in PBS and mounted on glass coverslips. Nuclei were stained with TO-PRO-3 Iodide (642/661, Thermofisher Scientific Invitrogen, T3605). Negative controls were performed in the absence of primary antibodies. Confocal image z-stacks were captured through 2 µm steps and used for double-labeled cell count, using ImageJ.

### 2.7. Drugs 

Drugs CGP55845 5 µM (Abcam, Cambridge, UK); CYN 154806 1 µM (Tocris, Bristol, UK), LY-367385 100 µM (Abcam, Cambridge, UK), and DPCPX 1 µM (Sigma Aldrich, Milano, IT, USA) were bath-applied. 

### 2.8. Software

Data analysis was performed with Clampfit 10, Origin 8.0 (Microcal Software, ImageJ 1.51n), Microsoft Office, ImageJ (NIH) and MATLAB 7.6.0 R2008A (Mathworks, Natick, MA, USA). Schematic representations were created with BioRender.com (accessed on 1 March 2022).

### 2.9. Statistical Analysis

Calcium data were presented as box and whiskers plots. Each box was defined by the 25th and 75th percentile; the central line indicated the median and the dot indicated the mean value. The whiskers represented the minimum and maximum values in one and a half times the interquartile range. To identify the conditions (S1, R1, S2, and R2) that differ from the basal condition (bsl), depending on the normality of the data, we used one-way repeated measures ANOVA with a post hoc Holm-Sidak test or one-way repeated measures ANOVA on ranks with Dunnett’s test. For electrophysiological experiments, data were presented as mean ± standard error of the mean (SEM). Individual values were also reported. Values before and after SST interneuron stimulation were compared using either a parametric paired *t*-test or a nonparametric Wilcoxon signed-rank test, depending on the normality of the data. Statistical differences were established, with *p* < 0.05 (*), *p* < 0.01 (**) and *p* < 0.001 (***). When indicated, data were compared with values obtained in SST-Cre mice in the absence of antagonists (*t*-test, (#) *p* < 0.05).

## 3. Results

### 3.1. Astrocyte Response to SST Interneurons in the VCx

To investigate whether SST interneuron activation in the VCx evokes a Ca^2+^ response in astrocytes, we used an AAV-based strategy (Figure 1A–C) to induce the expression of the light-gated cation channel Channelrhodopsin-2 (ChR2) specifically in SST interneurons (Figure 1D,E) and the genetically encoded Ca^2+^ indicator GCaMP6f specifically in astrocytes (Figure 1F,G). Two weeks after AAV injections of SST-Cre mice, we performed optogenetic activation of SST interneurons and 2-photon Ca^2+^ imaging experiments in coronal brain slices containing the VCx. We applied two successive 30 light pulses (λ = 470 nm, 150 ms pulse duration, 1 Hz, 3–4 mWs) that induced an intense firing discharge in ChR2-expressing SST interneurons (Figure 1B) and monitored Ca^2+^ activity in GCaMP6f-expressing astrocytes (Figure 2A–C).

We found that both the frequency of astrocytic Ca^2+^ microdomain events and the number of active microdomains were significantly increased immediately after SST interneuron stimulation and remained elevated during the recovery time (Figure 3A). In contrast, in proximal processes the frequency of Ca^2+^ events and the number of active processes showed a tendency to increase immediately after the first stimulation and returned to basal levels during the recovery time (Figure 3B). This transient increase was even more evident after the second stimulation, when it reached statistical significance. These results suggest that astrocyte thin processes respond remarkably to SST interneuron activation, while the response observed at proximal processes is less intense and recovery is faster with respect to microdomain response.

### 3.2. Astrocyte Response to SST Interneuron Stimulation Is Mediated by GABA_B_R-IP_3_R2 Pathway Activation

We next investigated whether the astrocyte response to SST interneuron stimulation is mediated by GABA_B_R activation. We found that in the presence of GABA_B_R selective antagonist CGP 55845 (5 µM), the frequency of events and overall number of active microdomains and proximal processes did not increase after SST interneuron activation (Figure 4A–C). Surprisingly, a significant reduction in the frequency of events at microdomains and the number of active microdomains was observed during the recovery times (Figure 4B). At present, we are unable to provide a plausible mechanistic hypothesis that could explain this reduction; additional experiments are needed to clarify this issue. Notwithstanding this observation, the absence of an increase in the Ca^2+^ activity in the different compartments suggests that in VCx the astrocyte response to an intense SST interneuron stimulation is mediated by the activation of GABA_B_Rs, probably expressed at the astrocytic plasma membrane.

GABA_B_R activation in astrocytes mediates Ca^2+^ elevations through an IP_3_R2-dependent mechanism [14]. To further support the hypothesis that the GABA_B_R-IP_3_R2 pathway is involved in the mechanism of astrocytic Ca^2+^ increases upon an intense stimulation of SST interneurons in the VCx, we generated IP_3_R2KO::SST-Cre mice (see Section 2). In slices from IP_3_R2KO::SST-Cre mice, we found that 30 light pulse stimulation of SST interneurons failed to induce significant astrocyte Ca^2+^ increases (Figure 5A,B). These data indicate that the GABA_B_R-IP_3_R2-mediated signaling pathway plays a crucial role in the astrocyte response to SST interneuron stimulation.

### 3.3. SST Contribution to Astrocytic Ca^2+^ Response upon SST Interneuron Stimulation 

We next explored whether the release of somatostatin by SST interneurons is involved in the astrocytic Ca^2+^ response, as we previously reported in the SSCx [27]. With this aim, we performed experiments in the presence of the specific SSTR antagonist CYN 15406 (1 µM). We observed that in the presence of CYN15406, SST interneuron activation induced a significant increase in the overall number of active microdomains, while the increase in the Ca^2+^ event frequency at microdomains, immediately after the SST stimulation did not reach statistical significance (Figure 6A–C). The effect of CYN 15406 was even more pronounced at the level of the proximal processes, where no statistically significant responses were observed after SST interneuron stimulation (Figure 6B). Taken together, these results suggest that the neuropeptide SST contributes to the Ca^2+^ response in astrocytes evoked by SST interneuron stimulation.

### 3.4. Intense SST Interneuron Stimulation Evokes a Disinhibition of PNs by Reducing the Amplitude of Inhibitory Currentswe Next Investigated Whether the Recruitment of Astrocytes after an Intense SST Interneuron Stimulation May Modulate Synaptic Inhibition onto PNs 

To address this hypothesis, we performed whole-cell patch-clamp recordings from layer II/III PNs in VCx slices from SST-Cre mice (see Section 2). Inhibitory post-synaptic currents (IPSCs) were evoked in PNs by brief light pulses (λ = 470 nm, 4 ms duration) stimulus intensity adjusted to evoke 50% maximal IPSC amplitude (Appendix A), applied at 0.1 Hz for 10 min, and grouped in time bins of 2.5 min to plot the time course of IPSC amplitude. Then, we applied the protocol of intense SST interneuron stimulation (30 light pulses, 150 ms duration, 1 Hz, 3–4 mW) that evoked the astrocyte Ca^2+^ response; after resuming IPSC recordings for another 10 min, we applied a second SST interneuron stimulation as in the Ca^2+^ imaging experiments (Figure 7A).

The mean IPSC amplitude from PNs was significantly decreased following both the first and the second SST interneuron stimulations (Figure 7B–D). This result suggests that an intense SST interneuron stimulation is followed by a disinhibition of PNs.

### 3.5. GABA_B_R-IP_3_R2 Pathway Involvement in the Modulation of IPSC Amplitude of PNs

To understand whether astrocytes are involved in the disinhibition of PNs after SST interneuron hyperactivity, we performed experiments in the presence of GABA_B_ antagonist CGP 55845, which blocks astrocytic Ca^2+^ responses. We observed that in the presence of CGP 55845 (5 µM), the decrease in the IPSCs amplitude was significantly larger with respect to control (Figure 8A–C). These results suggested that in the absence of astrocyte recruitment, the disinhibition was not prevented, and instead a stronger disinhibition was observed, suggesting that SST interneuron to astrocyte signaling counterbalances the disinhibition of PNs.

To support our hypothesis that IP_3_R2-mediated astrocyte Ca^2+^ signaling plays a crucial role in reducing the disinhibition of PNs evoked by intense SST interneuron stimulation, we performed experiments in the IP_3_R2KO::SST-Cre mice. Similar to the data obtained after blocking GABA_B_Rs, in the IP_3_R2KO::SST-Cre mice the disinhibition after SST interneuron stimulation was stronger with respect to controls (Figure 9A–C).

### 3.6. SST Receptors Participate to the Mechanism of Disinhibition of PNs

We next explored the involvement of SST receptors in the modulation of the IPSC amplitude on PNs. We found that in the presence of the SSTRs antagonist CYN 154806 (1 µM) the disinhibition of PNs was almost completely impaired after the first stimulation and only a small and not significant disinhibition was observed after the second stimulation (Figure 10A–C). These data suggest that activation of neuronal SST receptors triggers a strong reduction of the inhibitory currents, to control an excessive action of SST interneuron signaling.

### 3.7. Astrocyte Activation by an Intense Stimulation of SST Interneurons Induces the Release of ATP/Adenosine That Modulates Inhibitory Transmission

The release of astrocytic glutamate has been extensively shown to modulate excitatory synaptic transmission through the activation of presynaptic metabotropic mGluR1 [10,33,34,35,36]. Therefore, we investigated whether the gliotransmitter glutamate could contribute to the modulation of SST interneurons signaling to PNs. We found that in the presence of the mGluR1 antagonist LY 367385 (100 µM), the disinhibition of PNs triggered by an intense stimulation of SST interneurons was unchanged (Figure 11A–C). Adenosine acts on either pre- or postsynaptic adenosine receptors, decreasing or enhancing synaptic transmission [18,37,38]. We then investigated the possible involvement of astrocytic ATP release, which is rapidly degraded to adenosine by membrane-bound ecto-nucleotidases. Thus, we performed experiments in the presence of the selective antagonist of A_1_Rs DPCPX (1 µM) and found that after both the first and the second SST interneuron stimulation, the IPSC amplitudes were further decreased (Figure 11D–F), suggesting that activation of A_1_R favors an increase in IPSC amplitude that counterbalances the disinhibition of PNs.

If our hypothesis is correct, the A_1_R antagonist DPCPX should not impair the astrocyte Ca^2+^ response to sustained SST interneuron stimulation. In the presence of DPCPX, we observed a significant increase in the frequency of Ca^2+^ events and in the number of active ROIs at the microdomains (Figure 12A). At the proximal processes (Figure 12B), the increases observed in terms of the frequency of Ca^2+^ events and in the number of active ROIs were appreciable, though not statistically significant, similar to the frequencies observed in the absence of an antagonist. Therefore, the greatly preserved astrocyte response in the presence of DPCPX suggests that A_1_R activation plays a crucial role in the modulation of SST interneuron synaptic transmission downstream of astrocyte Ca^2+^ signals.

## 4. Discussion

In this study, we report that astrocytes recruited by SST interneuron signaling contribute to the modulation of the inhibitory synaptic transmission by this class of interneurons to VCx layer II/III PNs. We observed that the strength of the inhibitory input to PNs is significantly reduced following a short period of SST interneuron hyperactivity. This disinhibition is, however, significantly counterbalanced by astrocytes that are activated by SST interneuron signaling. At the basis of this mechanism, we propose the following sequence of events (Figure 13): SST interneuron stimulation releases, in addition to GABA, the neuropeptide SST that induces a reduction of IPSC amplitude in PNs favoring a disinhibition of these cells. Simultaneous activation of GABA_B_Rs in astrocytes triggers Ca^2+^ elevations, modulated by SSTRs, and subsequent ATP/adenosine release. Then, A_1_R activation by astrocytic adenosine favors an increase in IPSC amplitude in PNs that counterbalances the disinhibition of PNs. Taken together, our results suggest that after an intense activity of VCx SST interneurons the degree of inhibitory transmission to PNs is the result of dynamic interactions between the interneuron and astrocytes. 

In the hippocampus, SST interneuron stimulation evokes an astrocyte Ca^2+^ response that is predominantly mediated by the activity of the astrocyte transporter GABA transporter-3 (GAT-3) and only partially by the activation of GABA_B_Rs [39]. In contrast, in the VCx, the Ca^2+^ response to SST interneuron hyperactivity depends on the activation of GABA_B_Rs, similar to the astrocyte Ca^2+^ response that we previously observed in the SSCx [27]. In both cortical regions, the release of somatostatin by SST interneurons appears to have a role in the modulation of GABA_B_-mediated Ca^2+^ response in astrocytes.

Recently, the use of GECIs has greatly improved our analysis of the complexity of astrocytic Ca^2+^ signal. Indeed, in our study with the GECI CGaMP6f, we found different dynamics of Ca^2+^ responses at different astrocytic compartments. While the Ca^2+^ responses at the microdomains remain elevated during the recovery time, at the proximal processes we observed a less significant increase of astrocyte Ca^2+^ activity immediately after the 30 light pulse stimulation that returns to basal levels during the recovery time. These results suggest that the dynamics of Ca^2+^ activity at the microdomains and proximal processes are differently regulated. This difference could be explained by the fact that microdomains, due to their close localization to synaptic elements, can sense neuronal activity in a more direct and long-lasting manner than the proximal processes in which the Ca^2+^ responses are the result of an integration of the activity present in different microdomains and/or the activation of receptors by the spillover of neurotransmitters from distant synapses. Our results suggest that somatostatin released by SST interneurons also participates in the mechanism of astrocyte Ca^2+^ dynamics by favoring a sustained microdomain response. In proximal processes, the contribution of somatostatin to astrocyte response is even higher because the response is completely abolished after the SSTR blockade. This stronger effect of the SSTR antagonist on proximal process responses, compared to responses observed at microdomains, indirectly supports the hypothesis that the level of activity in astrocytic fine processes is subsequently integrated in the proximal processes. According to this hypothesis, manipulations that reduce microdomain responses could more efficiently impair proximal process responses. 

In our study, we observed that the intense stimulation of SST interneurons triggers a long-lasting reduction of inhibitory transmission to PNs that is partially counterbalanced by GABA_B_R-IP_3_R2-mediated astrocyte activation. This disinhibition is, indeed, strengthened in the presence of the specific antagonist of the GABA_B_R that blocks the astrocytic Ca^2+^ response to SST interneurons, and in IP_3_R2::SST-Cre mice, where the astrocytic Ca^2+^ response to SST interneurons is impaired. Because neuronal GABA_B_Rs mediate inhibitory effects, we ruled out the possibility that the stronger disinhibition observed in the presence of the GABA_B_R antagonist with respect to controls could be due to inhibition of neuronal GABA_B_Rs.

Somatostatin, like other neuropeptides, is stored in dense-core vesicles that requires high frequency firing activity to be released [40]. The effects of neuropeptides last longer with respect to classical neurotransmitters because neuropeptide receptors are often distant from the release site and there are no selective re-uptake mechanisms that could restrict their diffusion [40]. In SST interneuron cortical circuits, somatostatin acts as a co-transmitter with GABA that exerts its modulatory effect pre- or postsynaptically, supporting a fine-tuning of neuronal signaling [40]. For example, in the sensory thalamus, GABA receptor-mediated transmission is inhibited by SST via a pre-synaptic mechanism [41] and in VCx, SST presynaptically suppresses excitatory inputs to the PV interneurons, enhancing visual processing and perception [28]. Postsynaptic SSTR activation results most frequently in the activation of K^+^ channels [42] that leads to postsynaptic slow, long-lasting hyperpolarization [40]. Our experiments revealed that after inhibiting SSTRs, the disinhibition of PNs triggered by an intense stimulation of SST interneurons was almost completely lost. We speculate that the activation of SSTRs during the intense SST interneuron stimulation is involved in the reduction of the inhibitory currents that counteracts the excessive inhibitory signals. Ultrastructural studies aimed at identifying the specific localization of SSTRs in different subtypes of interneurons, including SST interneurons, will provide support for this hypothesis.

Adenosine A_1_Rs are extensively distributed in the brain, with notable abundance in the cortex, hippocampus, and cerebellum [43]; they are heterogeneously expressed in pre- and postsynaptic membranes [44,45,46]. Pre-synaptic A_1_Rs inhibit neurotransmitter release through G-protein-coupled inhibition of voltage-dependent Ca^2+^ channels [47], while postsynaptic A_1_Rs induce neuronal hyperpolarization through activation of inwardly rectifying K^+^ channels, which regulates burst firing activity [48,49,50]. In contrast, Matos et al. recently described that CA1 hippocampal astrocytes sense GABA released by moderate SST interneuron activity via GAT-3-mediated Ca^2+^ elevations, and release ATP that is rapidly converted to adenosine. The following activation of A_1_Rs, which authors hypothesize to be present postsynaptically, results in a potentiation of SST interneuron inhibitory transmission to PNs [39]. The increase in IPSC amplitude by postsynaptic A_1_R activation could be explained by the mechanism linked to the cyclic adenosine monophosphate (cAMP) pathway. A_1_Rs are GPCRs coupled to the subunit Gi, which inhibits the cAMP pathway by inhibiting adenylate cyclase activity. In turn, the decrease in the production of cAMP results in a decreased activity of PKA and increased postsynaptic GABA_A_R function due to a reduced PKA-mediated negative modulation of the β subunit [51]. A similar mechanism was suggested for the enhancement of GAB_A_A receptor-mediated IPSC amplitude in CA1 pyramidal neurons following ischemia [38].

Consistent with this mechanism, in the presence of a specific A_1_R antagonist, we observed a significant decrease of PN IPSC amplitudes after SST interneuron stimulation. In summary, we proposed that in the VCx, astrocytes counterbalance the disinhibition of PNs by releasing ATP/adenosine, which through activation of postsynaptic A_1_R induce a decrease in the activity of PKA, which in turn limits the phosphorylation of GABA_A_ receptors enhancing GABAergic neurotransmission. To support this hypothesis, additional experiments aimed at providing ultrastructural evidence for the presence of A_1_Rs in PNs of the VCx are needed.

At the basis of sensory information processing and behavioral responses to an ever-changing environment is the high plasticity of excitatory and inhibitory synaptic transmission. Astrocytes are functionally integrated in neuronal circuits to provide a further level of complexity in the modulation of synaptic circuitry. Here, we showed that an intense stimulation of SST interneurons generates two opposing mechanisms that shape the inhibitory transmission of SST interneurons onto PNs: (i) a neuronal mechanism that, through SST release, decreases GABA currents; and (ii) a GABA_B_-mediated astrocytic mechanism that, through ATP/adenosine release and activation of A_1_ receptors, favors the inhibitory tone. The preferred postsynaptic targets of SST interneurons in layer II/III are PNs [26]. Layer II/III and layer V SST interneurons receive excitatory inputs from neighboring PNs and from cortico-cortical and thalamic afferents [26]. When SST interneurons are activated by neighboring PNs, their function is to provide feedback inhibition onto the same PNs or to inhibit neighboring PNs to increase the contrast of competing inputs [26]. In line with this latter idea, SST interneurons contribute to surround suppression in the primary VCx [26]. In contrast, when SST interneurons are activated by cortico-cortical and thalamic afferents, they also provide feedforward inhibition to PNs [26]. Feedback or feedforward inhibition onto PNs is dependent on the firing frequency in the presynaptic afferents, i.e., SST interneurons act as rate detectors, providing a frequency filter to the postsynaptic PNs [26]. During sensory information processing, visual habituation, defined as a loss of cortical response to a continued exposure to a given stimulus, is accompanied by an increased activity of SST interneurons and a reduced activity of PV interneurons [26]. In light of our results, it would be of great interest to understand whether astrocytes participate in visual stimulus processing by finely shaping phenomena such as the surround inhibition and the adaptation. In the hippocampus, the interneuron to astrocyte signaling plays a role in complex mechanisms of heterosynaptic modulation that potentiate or inhibit excitatory transmission [10,13]. In our study of the VCx, we limited our analysis of the SST interneuron-to-astrocyte signaling to the effect of astrocyte recruitment to inhibitory circuits. However, one may speculate that activation of astrocytes by SST interneuron activity can also influence unrelated heterosynaptic excitatory transmission with unpredictable consequences in brain function.

Given the abundance of the roles played by astrocytes in brain function, it is not surprising that increasing evidence supports the involvement of astrocyte signaling in neurological disorders. Accordingly, alterations of interneuron-astrocyte reciprocal interactions may be also involved. For example, activation of astrocytes by GABAergic signaling may be important in brain disorders, such as epilepsy [52]. In the cortex, the absence of synaptic inhibition is not sufficient to produce epileptic activity. However, disinhibition can play an important role in epilepsy, as it can unveil an abnormal excitatory activity and lower the threshold for triggering self-sustained seizures [52]. In this context, astrocytes may play an anticonvulsant role by counteracting disinhibition, while a defective interneuron-to-astrocyte signaling could favor epileptic activity.

In conclusion, the main goal of the present study was to gain further insights into the reciprocal signaling between GABAergic interneurons and astrocytes. We described a complex mechanism of reciprocal functional interactions between astrocytes and SST interneurons that controls the inhibitory tone in VCx circuits. 

## Figures and Tables

**Figure 1 cells-11-01400-f001:**
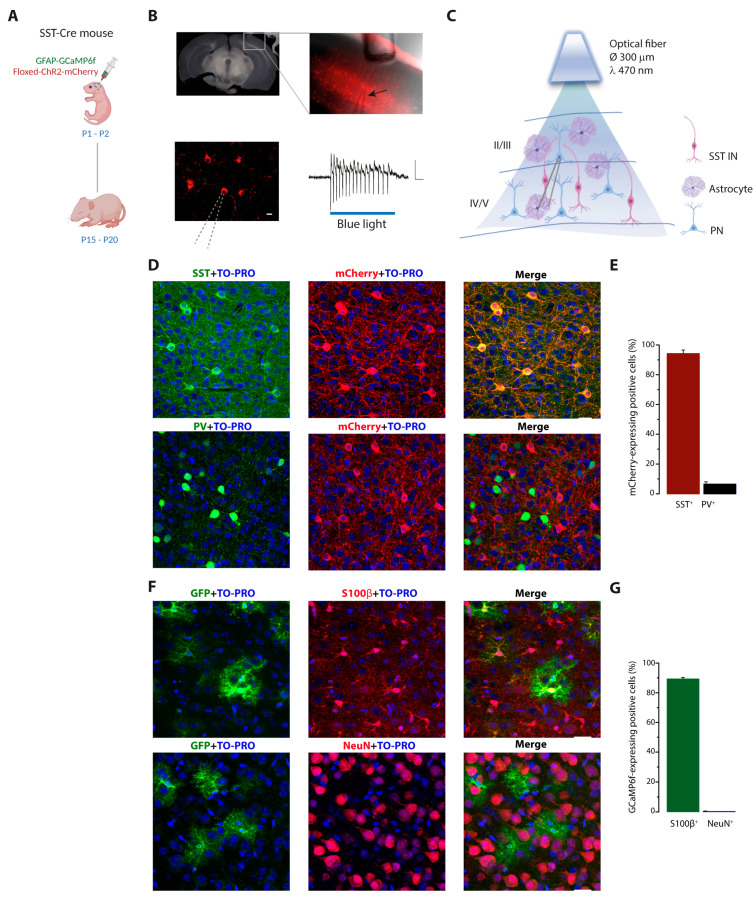
Experimental approach for Channelrhodopsin-2 (ChR2)-mCherry specific expression in somatostatin (SST) interneurons and GCaMP6f specific expression in astrocytes from the visual cortex (VCx). (**A**) Schematic representation of adeno-associated virus (AAV) injections in the VCx of newborn SST Cre mice and brain slice preparations at least 2 weeks after injections. (**B**) Left upper panel, representative differential interference contrast (DIC) image of a coronal slice containing the VCx (square). Right upper panel, combined fluorescence and DIC images of the square shown in left panel, showing ChR2-mCherry expression in VCx SST interneurons, the optical fiber for optogenetic stimulation, and the patch pipette (arrow) for electrophysiological recordings. Scale bar, 50 µm. Bottom, fluorescence image of ChR2-mCherry-expressing SST interneurons (scale bar, 10 µm) and a representative action potential discharge, recorded in cell-attached mode, upon a single 150 ms blue light (470 nm) pulse stimulation. Scale bars, 20 ms, 20 pA. (**C**) Schematic of the experimental approach. Ca^2+^ imaging in layer II/III astrocytes and whole-cell patch-clamp recordings from layer II/III pyramidal neurons (PNs) were performed before and after optogenetic stimulation of SST interneurons. In this figure and the other figures, violet-shaped cells represent astrocytes, red cells represent SST interneurons, and blue cells represent PNs. (**D**) Confocal microscope fluorescence images of the VCx from a mouse injected with AAV2/1.EF1.dflox.hChR2(H134R)-mCherry.WPRE.hGH, illustrating the specific localization of ChR2-mCherry in SST interneurons. Red fluorescence, ChR2-mCherry (α-RFP staining); blue fluorescence, nuclear TO—PRO-3; specific green staining for either SST interneurons (α-SST staining) or parvalbumin (PV) interneurons (α-PV staining). Scale bar, 25 μm. (**E**) Bar chart showing that cells expressing ChR2-mCherry are mainly SST interneurons. α-SST: 406 SST+ cells out of 432 ChR2-mCherry+ cells (3 mice, 8 slices); α-PV: 33 PV+ cells out of 461 ChR2-mCherry+ cells (3 mice, 6 slices). (**F**) Confocal microscope fluorescence images of the VCx from a mouse injected with AAV5.GfaABC1DcytoGCaMP6f.SV40, illustrating the specific localization of GCaMP6f in astrocytes. Green fluorescence, GCaMP6f (α-GFP staining); blue fluorescence, nuclear TO-PRO-3; specific red staining for either astrocytes (α-S100β staining) or neurons (α-NeuN staining). Scale bar, 25 μm. (**G**) Bar chart showing that cells expressing GCaMP6f are mainly astrocytes. α-S100β: 2528 S100β+ cells out of 2856 GCaMP6f-GFP+ cells (5 mice, 12 slices); α-NeuN: 2 NeuN+ cells out of 510 GCaMP6f-GFP+ cells (2 mice, 6 slices).

**Figure 2 cells-11-01400-f002:**
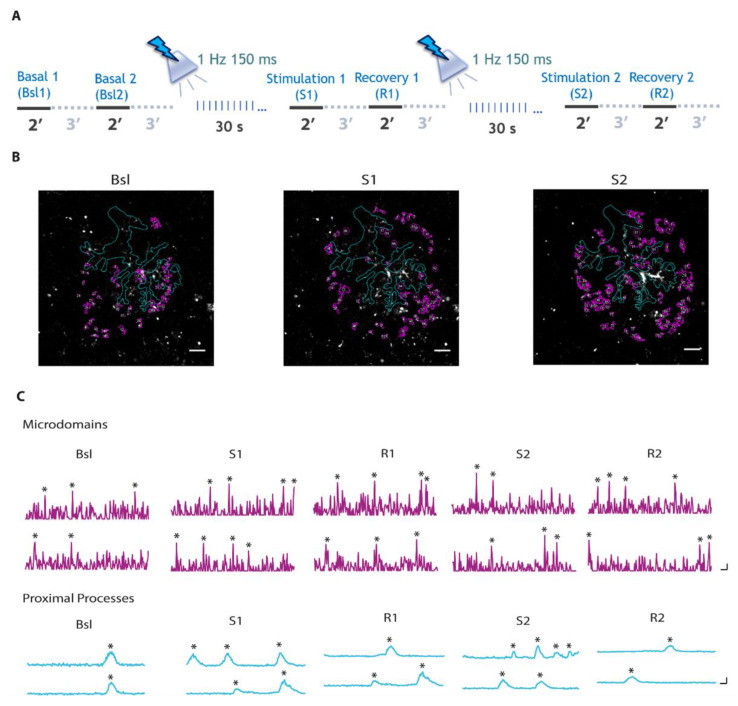
Combined Ca^2+^ imaging in astrocytes and optogenetic stimulation of SST interneurons. (**A**) Calcium imaging 2 min recording protocol in astrocytes before and after optogenetic stimulation of SST interneurons with two successive 30 light pulse stimulations. (**B**) 2-photon fluorescence images of a representative GCaMP6f-expressing astrocyte in layer II/III VCx from a SST-Cre mouse at different time points. ROIs defined by GECIquant plugin of Image J (see Section 2 for details) are shown in violet for microdomains. In cyan, astrocytic main structure reconstruction from t-series maximal projection. Scale bar, 10 μm. (**C**) Representative Ca^2+^ signal dynamics at different astrocytic compartments before and after successive 30 light pulses. * indicate significant Ca^2+^ peaks detected in a semi-automated manner by a MATLAB script [27]. Scale bars, 10 s, 2 ΔF/F_0_.

**Figure 3 cells-11-01400-f003:**
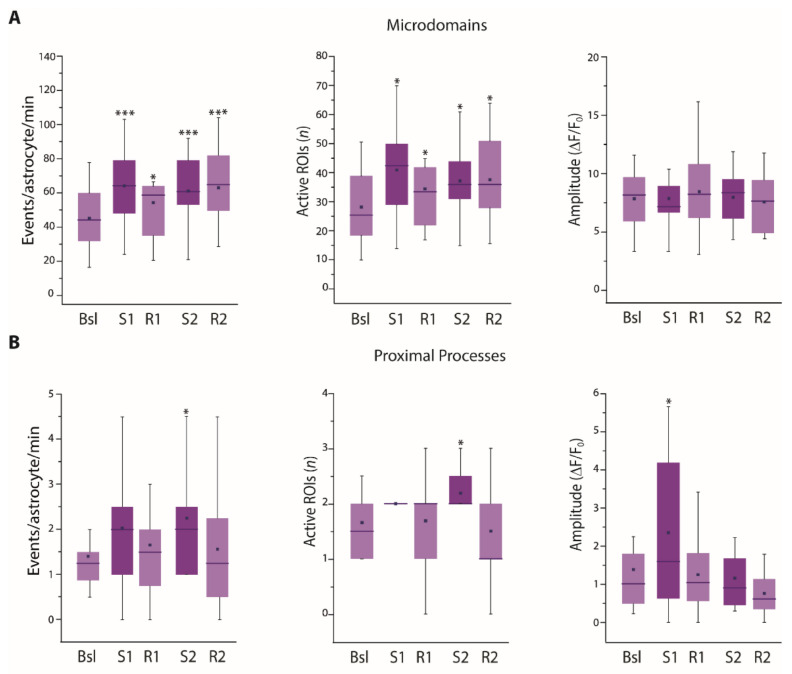
Astrocyte Ca^2+^ response upon successive 30 light pulse SST interneuron stimulations. (**A**,**B**) Average data of Ca^2+^ signal dynamics at microdomains (11,984 total events) (**A**) and proximal processes (330 total events) (**B**) of GCaMP6f-expressing astrocytes upon SST interneuron light pulse stimulation (18 astrocytes, 13 slices, 5 mice). * *p* < 0.05, *** *p* < 0.001, One-way repeated measures ANOVA with a post hoc Holm-Sidak test and one-way repeated measures ANOVA on ranks with Dunnett’s test were used to compare all other conditions to Bsl condition.

**Figure 4 cells-11-01400-f004:**
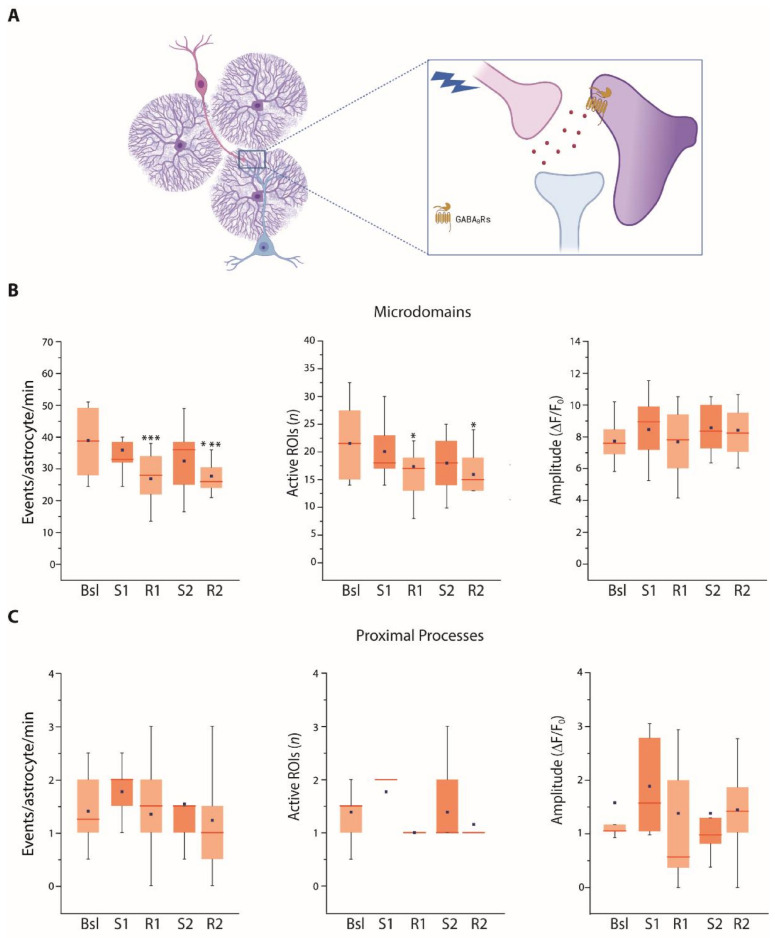
Astrocyte Ca^2+^ response to SST interneuron stimulation is mediated by GABA_B_Rs. (**A**) Scheme showing the hypothesis that VCx astrocyte Ca^2+^ response to SST interneuron stimulation is mediated by GABA_B_R activation. (**B**) Average data of Ca^2+^ signal dynamics at microdomains (5430 total events) in the presence of the GABA_B_R antagonist CGP 55845 (13 astrocytes, 8 slices, 4 mice). (**C**) Average data of Ca^2+^ signal dynamics at proximal processes (234 total events) in the presence of CGP 55845 (13 astrocytes, 8 slices, 4 mice). * *p* < 0.05, *** *p* < 0.001, One-way repeated measures ANOVA with a post hoc Holm-Sidak test and one-way repeated measures ANOVA on ranks with Dunnett’s test were used to compare all other conditions to Bsl condition.

**Figure 5 cells-11-01400-f005:**
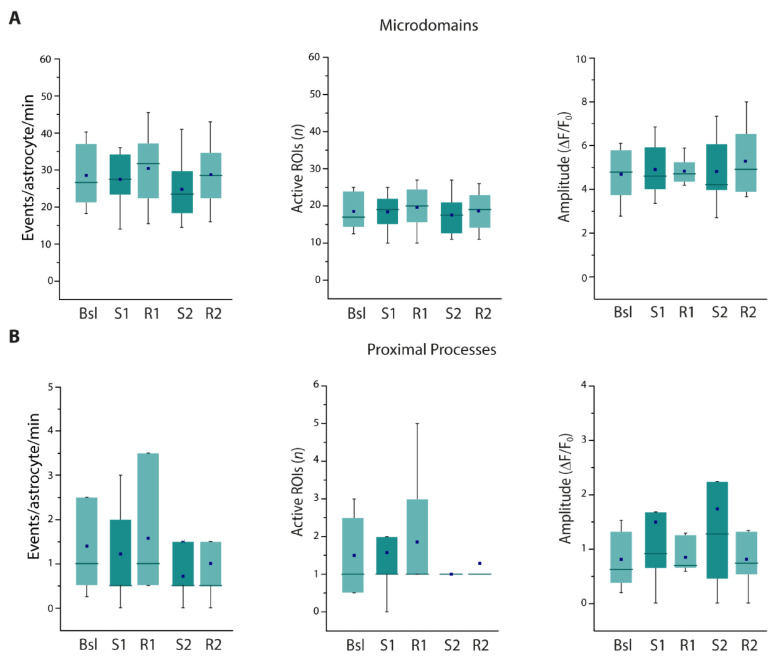
Astrocyte Ca^2+^ response to STT interneuron stimulation is impaired in IP_3_R2KO mice. (**A**,**B**) Average data of astrocytic Ca^2+^ signal dynamics at microdomains (2696 total events) (**A**) and proximal processes (102 total events) (**B**) before and after SST interneuron light pulse stimulation in IP_3_R2KO::SST-Cre mice (7–8 astrocytes, respectively; 6 slices, 3 mice).

**Figure 6 cells-11-01400-f006:**
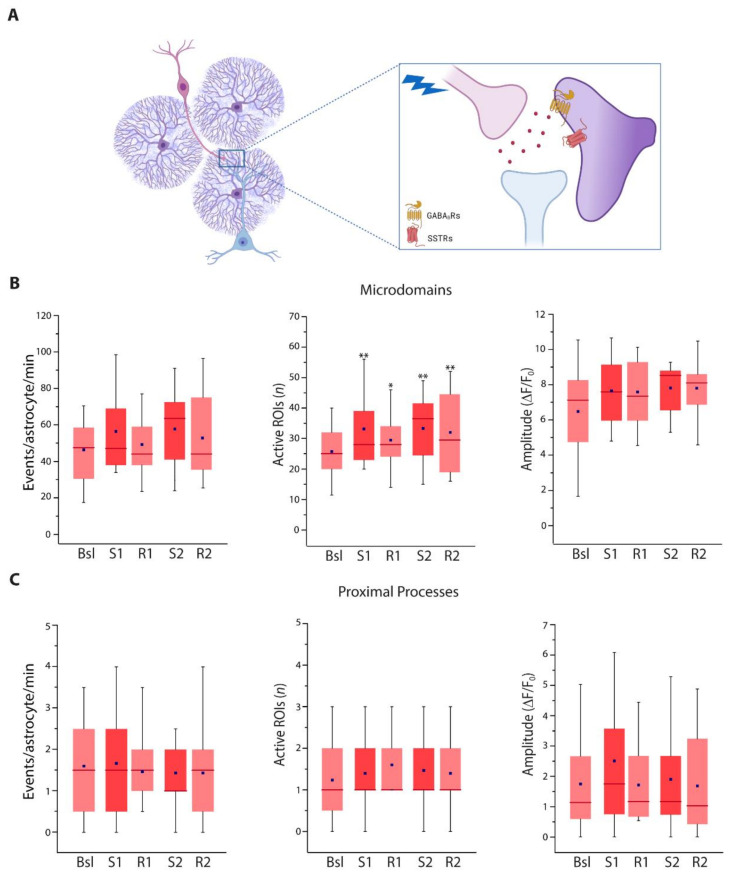
The neuropeptide somatostatin modulates the astrocyte Ca^2+^ response to SST interneuron stimulation. (**A**) Schematic showing the possible involvement of astrocyte SSTRs in the Ca^2+^ responses to SST interneuron stimulation. (**B**,**C**) Ca^2+^ signal dynamics at microdomains (7532 total events) (**B**) and proximal processes (228 total events) (**C**) in the presence of the SSTR antagonist CYN 15406 (15 astrocytes, 8 slices, 4 mice). * *p* < 0.05, ** *p* < 0.01, One-way repeated measures ANOVA on ranks with Dunnett’s test was used to compare all other conditions to Bsl condition.

**Figure 7 cells-11-01400-f007:**
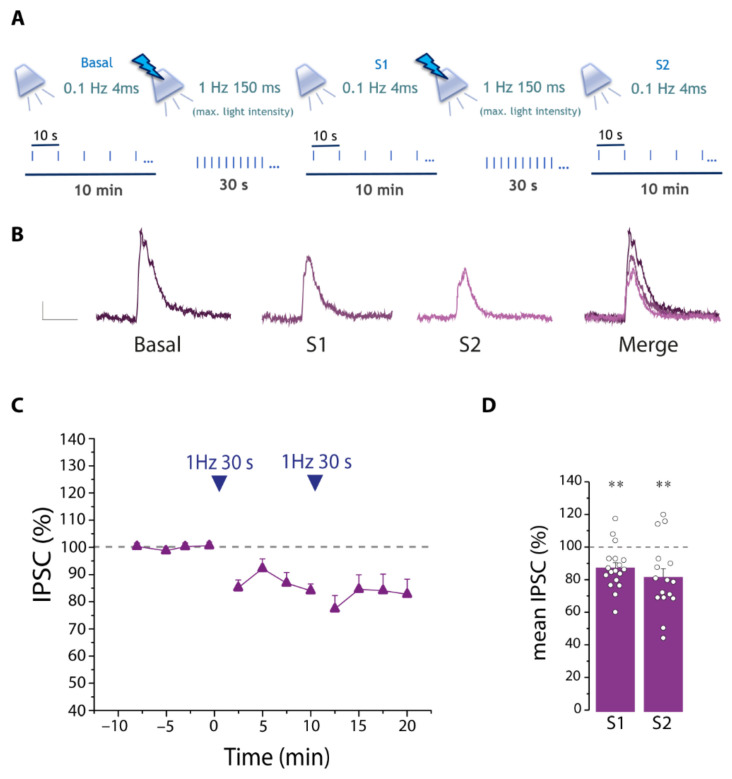
Intense stimulation of SST interneurons decreases the amplitude of inhibitory currents on PNs. (**A**) Schematic of the protocol for IPSC recordings in PNs and stimulation of SST interneurons. (**B**) Representative evoked IPSCs before (basal) and after the first (S1) and the second (S2) 30 light pulses and merger of all IPSCs. Scale bar, 50 ms, 10 pA. (**C**) Time course of the mean normalized IPSC amplitude before and after the 30 light pulse stimulations (*n* = 18). In this figure and the other figures reporting IPSCs amplitude vs time, t = 0 indicates the end of the first 30 pulse stimulation. (**D**) Quantification of the mean normalized IPSC amplitude during 10 min after the first (S1) and second (S2) stimulation. Note that an intense stimulation of SST interneurons disinhibits PNs. * statistical comparison with basal conditions, ** *p* < 0.01, paired *t*-test.

**Figure 8 cells-11-01400-f008:**
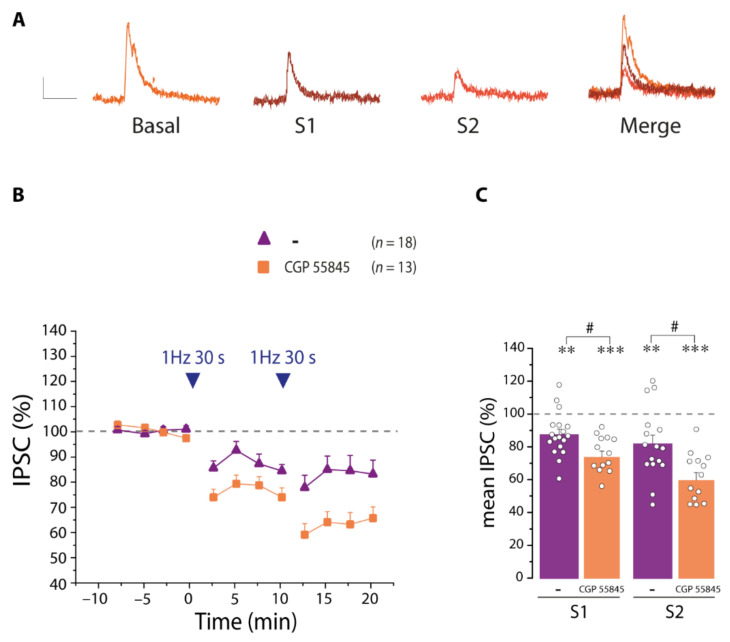
Blocking GABA_B_Rs induces a stronger disinhibition of PNs. (**A**) Representative IPSCs before (basal) and after the first (S1) and the second (S2) 30 light pulses and merger of all IPSCs, in the presence of the GABA_B_R antagonist CCP 55845. Scale bar, 50 ms, 10 pA. (**B**) Mean normalized IPSCs amplitude before and after the 30 light pulse stimulation in the presence of CGP 55845 (*n* = 13) (**C**) Quantification of the mean normalized IPSC amplitude during the 10 min after the first (S1) and second (S2) stimulation in the absence or presence of CGP55845. Note the increased disinhibition after blocking GABA_B_Rs. * statistical comparison with basal conditions, ** *p* < 0.01, *** *p* < 0.001, paired *t*-test. # statistical comparison in the absence and presence of antagonist, # *p* < 0.05, unpaired *t*-test. Note also that in this figure and the other figures reporting IPSC amplitudes in the presence of antagonists, the control values reported in Figure 7 have been plotted again to facilitate the comparison of PNs disinhibition in the absence and presence of antagonists.

**Figure 9 cells-11-01400-f009:**
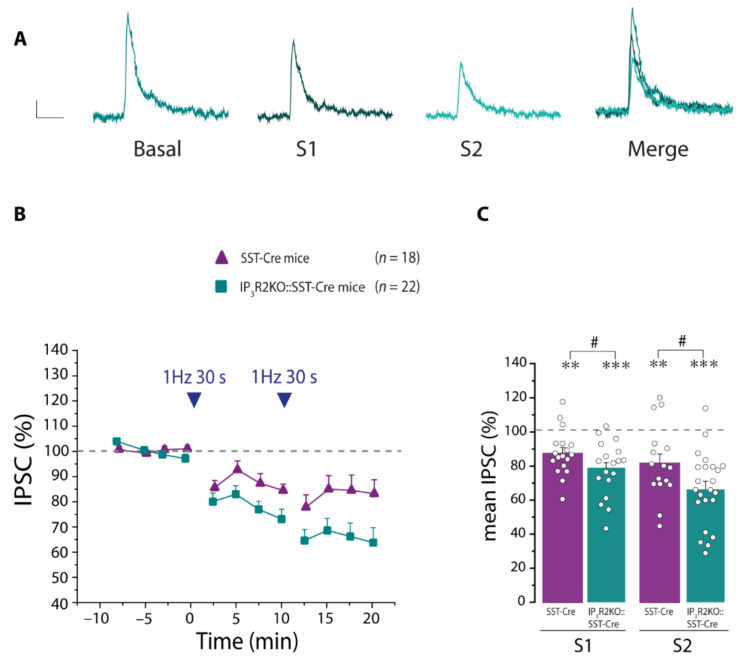
IP_3_R2-mediated signaling pathway counterbalances disinhibition of PNs. (**A**) Representative IPSCs before (basal) and after the first (S1) and the second (S2) 30 light pulses and merger of all IPSCs, in an IP_3_R2KO::SST-Cre mouse. Scale bar, 50 ms, 10 pA. (**B**) Mean normalized IPSCs amplitude before and after the 30 light pulse stimulation in the IP_3_R2KO::SST-Cre mice (*n* = 22). (**C**) Quantification of the mean normalized IPSC amplitude during the 10 min after the first (S1) and second (S2) stimulation in the IP_3_R2KO::SST-Cre mice. Note that in IP_3_R2KO::SST-Cre mice the disinhibition of PNs is stronger compared to that observed in SST-Cre mice. * statistical comparison with basal conditions, ** *p* < 0.01, *** *p* < 0.001, paired *t*-test. # statistical comparison with values in SST-Cre mice, # *p* < 0.05, unpaired *t*-test.

**Figure 10 cells-11-01400-f010:**
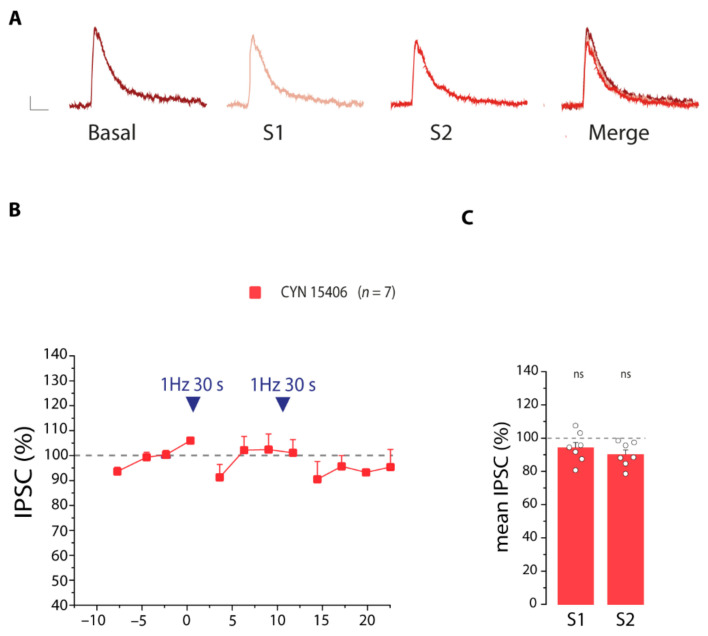
SST receptors participate to the mechanism of disinhibition of PNs. (**A**) Representative IPSCs recorded in the presence of the SSTRs antagonist CYN 15406, before (basal) and after the first (S1) and the second (S2) 30 light pulse stimulations. Scale bar 50 ms, 10 pA. (**B**) IPSCs amplitude before and after the 30 light pulse stimulation in the presence of the SSTR antagonist CYN 15406 (*n* = 7). (**C**) Quantification of the disinhibition of PNs after the first (S1) and second (S2) stimulations, in the absence and presence of CYN 15406. Note that in the presence of CYN 15406, an intense stimulation of SST interneurons does not decrease significantly the inhibitory currents on PNs. Statistical comparison with basal conditions, ns, not significant, paired *t*-test.

**Figure 11 cells-11-01400-f011:**
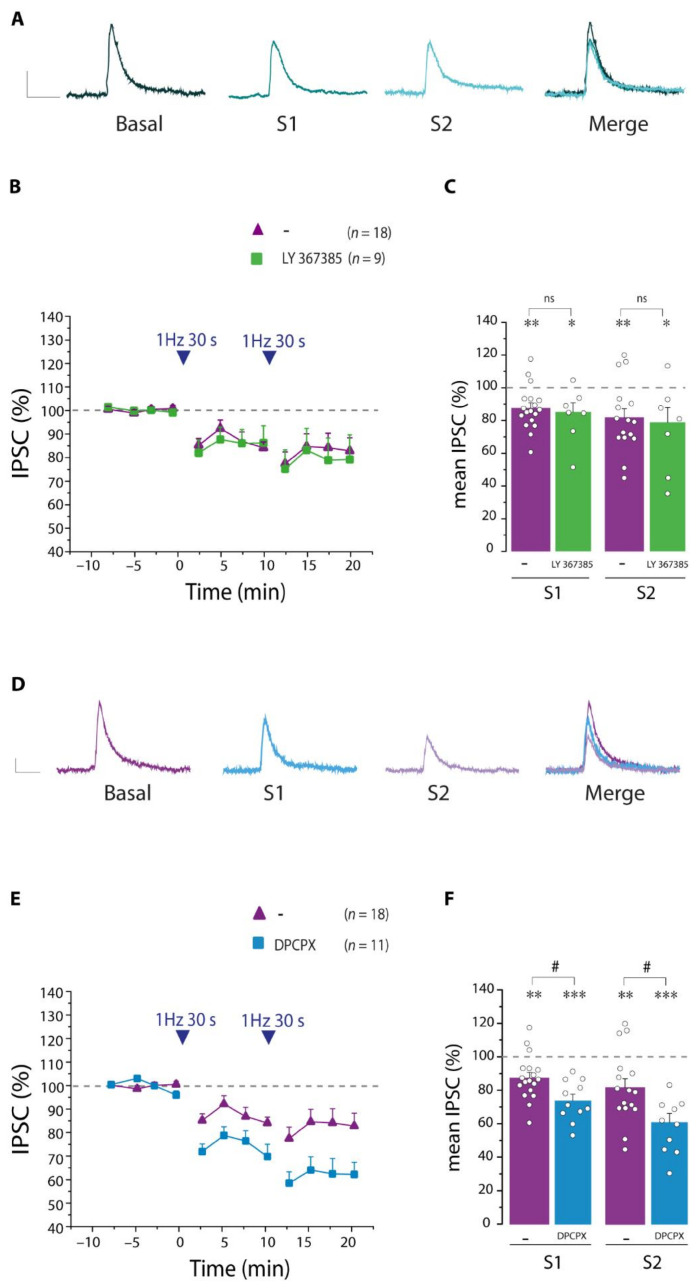
Activation of A_1_Rs enhances SST interneurons inhibitory transmission after intense SST interneuron stimulation. (**A**) Representative IPSC traces before (basal) and after the first (S1) and the second (S2) 30 light pulse stimulations in the presence of the mGluR1 antagonist LY 367385 and merge of all representative traces. (**B**) Normalized IPSC amplitudes before and after the 30 light pulse stimulations in the presence of LY 367385 (*n* = 9). (**C**) Quantification of normalized IPSC amplitude after the first (S1) and second (S2) stimulations in the absence and presence of LY 367385. Note that blocking mGluR1s does not change the disinhibition of PNs. (**D**) Representative IPSC traces before (basal) and after the first (S1) and the second (S2) 30 light pulse stimulation in the presence of A_1_R antagonist DPCPX and merger of all representative traces. (**E**) Normalized IPSCs amplitude before and after the 30 light pulse stimulation in the presence of DPCPX (*n* = 11). (**F**) Quantification of normalized IPSC amplitude after S1 and S2 in the absence and presence of DPCPX. Note that blocking A_1_Rs increases the disinhibition of PNs. * statistical comparison with basal conditions, * *p* < 0.05, ** *p* < 0.01, *** *p* < 0.001, paired *t*-test. # statistical comparison in the absence and presence of antagonist, ns, not significant, # *p* < 0.05, unpaired *t*-test. Scale bar, 50 ms, 10 pA.

**Figure 12 cells-11-01400-f012:**
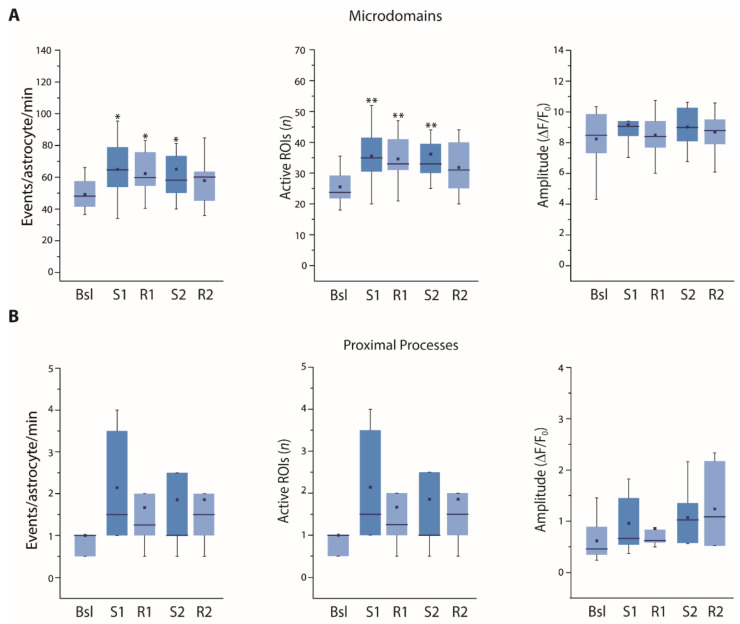
Astrocyte Ca^2+^ response to STT interneuron stimulation is highly preserved when A_1_Rs are blocked. (**A**,**B**) Microdomains (8350 total events) and proximal process (130 total events) Ca^2+^ responses to an intense stimulation of SST interneurons are highly preserved in the presence of the A_1_R antagonist DPCPX (12–7 astrocytes, respectively, 5 slices, 4 mice). * *p* < 0.05, ** *p* < 0.01, One-way repeated measures ANOVA with a post hoc Holm-Sidak test was used to compare all other conditions to Bsl condition.

**Figure 13 cells-11-01400-f013:**
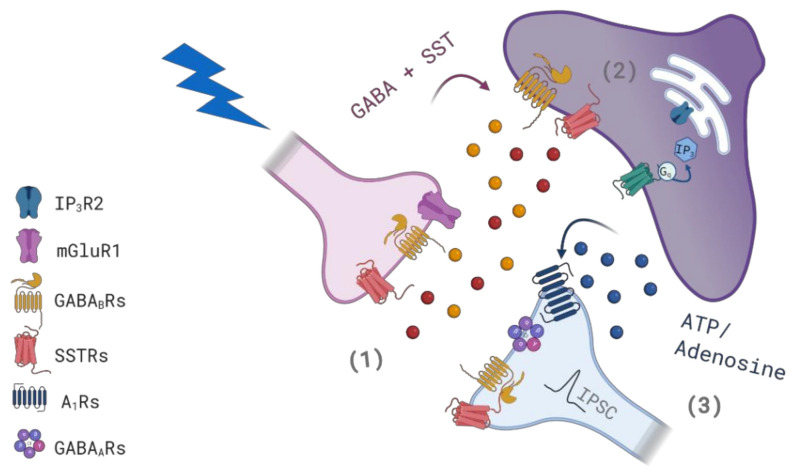
Schematic of the proposed cellular and molecular mechanism for IPSC amplitude modulation after SST interneuron hyperactivity. (1) An intense stimulation of SST interneurons in the VCx induces the release of GABA and somatostatin that, through activation of neuronal SSTRs, induces a reduction of IPSC amplitude in PNs, i.e., disinhibition of PNs. (2) Simultaneous activation of GABA_B_Rs in astrocytes triggers IP_3_R2-mediated Ca^2+^ elevations that are modulated by activation of astrocytic SSTRs. (3) Astrocytes release ATP/adenosine that by activating A_1_Rs, probably postsynaptic, favors an increase in IPSC amplitude that counterbalances the disinhibition of PNs.

## Data Availability

Data generated and/or analyzed during the current study are available from the corresponding author upon reasonable request.

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
