# Peer review of "Astrocytes Modulate Somatostatin Interneuron Signaling in the Visual Cortex"

_cells, 2022, doi:10.3390/cells11091400_

Round 1

Reviewer 1 Report

The new data and discussion added to this ms in the revision process have substantially increased the quality of the data present and, also, is showing evidence for the findings raised and clarified my previous concerns. The authors did address all the main points highlighted during revision round 1. The presented work is of high quality, and the combination of ephys and calcium measurements makes the work convincing and adds interesting and exciting results to the field.

Author Response

We thank the reviewer for his/her report.

Reviewer 2 Report

Jorge Henriques et al - Astrocytes Modulate Somatostatin Interneuron Signaling in the Visual Cortex

Jorge Henriques et al., demonstrate that astrocytes are actively participating in the regulation of somatostatin (SST) interneuron to pyramidal neuron (PN) communication, by counteracting the depression of inhibitory post-synaptic current (IPSC) in PN which follows intense SST interneurons activity. They confirm one of their previous report by showing that evoked activity of SST interneurons leads to an increase of intracellular calcium in astrocytes, which is mediated by GABAb receptors activation and modulated by the concomitant activation of SST receptors (SSTR). Next, they found that while activation of SSTR underlies the observed decrease in IPSC amplitude in PN following intense SST neurons activity, the same intense neuronal activity leads to the release of adenosine, which activates adenosine receptors 1 (A1R). The activation of neuronal A1R by (astrocytic) adenosine increases IPSC amplitude in PN, counteracting the effect of SSTR signaling. In slices from mice lacking the astrocyte-specific IP3R2 receptor, astrocytes become insensitive to evoked SST interneuron activity while the observed disinhibition at inhibitory synapses is stronger, clearly placing astrocytes as essential modulators of inhibitory transmission strength.

These results provide interesting insights into the mechanisms of astrocyte – inhibitory synapses interactions. This study further highlight the important roles of astrocytes in balancing synaptic activity, adding to the body of evidence now demonstrating these cells are important gears of brain circuits, but focuses on inhibitory synapses. These synapses have been understudied in the context of astrocyte-neuron interactions, compared to their excitatory counterparts. This specific focus is therefore a major strength of the authors’ work, and will be an important contribution to the field. The experiments are straightforward and executed following the highest methodological standards in the field. The display of results is logically structured, and the manuscript reads well.

This submission is a revision of a previous manuscript. I am satisfied by the new data the authors provided to answer the only major concern I had regarding the lack of astrocyte-specific manipulation. The observation that the lack of IP3R2 in astrocytes mimics the effects of both GABAb and A1 receptors antagonists on IPSC amplitude is now a major strength of the manuscript, clearly linking astrocytes calcium signaling with the modulation of disinhibition at SST - pyramidal neurons synapses. I also think the addition of immunohistochemistry data showing the cell-type specificity of ChR2-mCherry and GCaMP6f expression to SST interneurons and astrocytes, respectively, further strengthen the manuscript quality. The changes provided to the statistical analysis, main text and data display in figures is also improving the manuscript quality and its ease of comprehension by the reader. As for the few suggestions I made that the authors chose not to follow, I think they are minor issues that are not impinging the manuscript quality and conclusions, and are rather a matter of personal opinion.

I still have some minor comments, which are rather suggestions, that will be easily amended and should not impinge the publication of this manuscript in Cells and do not, in my opinion, require a further round of revision:

Material and methods:

  • Line 90-91: spelling mistake, Custom and not costume-made.
  • Section 2.2: Please indicate the titers of the AAV solutions injected.
  • Line 213-214: I think the author meant ‘to evoke 50% maximal IPSCs’ amplitude. I recommend the authors to check the whole manuscript for similar issues where the precision of which parameter of IPSCs the author speak of is lacking. This is in my opinion an important precision.
  • Line 236: I think the percentage of BS is missing? The authors could also precise the number and duration of washes, and the type of solvent they used (PBS?)
  • Line 241-242: “For controlling mCherry expressing neurons” I think the authors meant “for further controlling specificity of expression of ChR2-mCherry to somatostatin interneurons”.
  • Line 244: The species in which this anti-RFP antibody was raised is missing, I infer mouse from the secondary antibody used, but please correct.
  • Line 286: Did the author meant “when not indicated otherwise” instead of “when indicated”?

Results:

  • Line 351: I believe the author should write “statistical significance” and not “statistically significance”
  • Figure 3-6: The clarity of data presentation in these figures has clearly improved. I might just advise the author to maybe repeat in the legend, on top of what is clearly already said in the methods, that the post-hoc multiple comparisons are comparing all other conditions to the control, Bsl, condition for the sake of clarity.
  • Section 3.4 title: “by reducing inhibitory currents”: please add “amplitude” or “strength” at the end of the title.
  • Figure 7: as for Figure 3-6, I would add at the end of the legend that statistical comparisons are made by comparing the averaged amplitudes during ‘basal’ versus S1 and S2, even if already described in the methods. This is actually done in Figure 8-9 when writing : “*statistical comparison with basal conditions”. I think here the authors forgot to specify “*p<0.05” for paired t-tests. Also, the authors should specify that in these figures 8-9 that the # signs indicate use of unpaired t-tests, and not simply t-tests.
  • Line 469: instead of “astrocyte signaling” I would rather write “SST interneurons to astrocytes signaling”.
  • Line 531: “Adenosine receptor” should be plural, as there exist multiple adenosine receptors.
  • Figures are sometimes referred to as “Figure X” or “Fig. X”, the authors should choose one way according to editorial instructions.

DISCUSSION:

  • Figure 13, line 597: Could the authors check here and throughout the manuscript that they specify they only considered the amplitude of IPSC, by adding ‘amplitude’ after each mention of IPSC wherever relevant?
  • Line 611: pulse should be plural

Author Response

We thank the reviewer for his/her report. We changed the manuscript according to his/her comments.

Reviewer 3 Report

I think that this version is much better than first version.

Author Response

We thank the reviewer for his/her report.

This manuscript is a resubmission of an earlier submission. The following is a list of the peer review reports and author responses from that submission.

Round 1

Reviewer 1 Report

Please, see attached file.

Reviewer 2 Report

Henriques et al. - Astrocytes Modulate Somatostatin Interneuron Signaling in the Visual Cortex

Henriques et al., demonstrate that astrocytes are actively participating in the regulation of somatostatin (SST) interneuron to pyramidal neuron (PN) communication, by counteracting the depression of inhibitory post-synaptic current (IPSC) in PN that follows intense SST interneurons activity. They confirm a previous report by showing that evoked activity of SST interneurons leads to an increase of intracellular calcium in astrocytes, which is mediated by GABAb receptors activation and prolongated by the concomitant activation of SST receptors (SSTR). Next, they found that while activation of SSTR underlies the observed decrease in IPSC amplitude in PN following intense SST neurons activity, the same intense neuronal activity leads to the release of adenosine by astrocytes which activates adenosine receptors 1 (A1R). The activation of neuronal A1R increases IPSC amplitude in PN, counteracting the effect of SSTR signaling.

These results provide interesting insights into the mechanisms of astrocyte – inhibitory synapses interactions. This study further highlight the important roles of astrocytes in balancing synaptic activity, adding to the body of evidence now demonstrating these cells are important gears of brain circuits, but focuses on inhibitory synapses. These synapses have been understudied in the context of astrocyte-neurons interactions, compared to their excitatory counterparts. This specific focus is therefore a major strength of the authors’ work, and will be an important contribution to the field. The experiments are straightforward and executed following the highest methodological standards in the field. The display of results is logically structured, and the manuscript reads well, despite a lack of conciseness which could be improved. While I think this manuscript will be suitable for publication in Cells, I have one major and several minor concerns, that I would like the authors to address.

MAJOR CONCERNS

  1. I believe that there is a crucial gap in the results provided, which impinges on the main conclusion of the study. There are no astrocyte-specific manipulation. While I agree the literature is in favor of the authors hypothesis, with multiple evidence that increased calcium in astrocytes leads to the release of adenosine, I think the authors need to provide a proof that this is also the case in their specific model of astrocytes – SST interneurons – PN tripartite synapse. As it stands, all the conclusions regarding astrocyte-neurons interactions are purely correlative, due to the lack of cell type specificity of the pharmacological approaches used.

The best strategy would probably be to use an astrocyte specific, conditional knock out (cKO) of GABAb receptors, as was done by Mederos et al. (https://doi.org/10.1038/s41593-020-00752-x). With this model, the authors could test if in mice in which visual cortex (VCx) astrocytes lack the GABAb receptor, the calcium response of astrocytes to SST interneurons activation is lost, but crucially, if this also mimics the effect of GABAb and A1R antagonists on PN IPSC amplitude presented in Figure 7 and 9. Such results would undeniably place the astrocyte as the effector of A1R activation following SST interneurons intense firing. I understand the authors argument that the apparent lack of effect of A1R antagonist on astrocytes responses (Figure 10) indicates a mechanism downstream of astrocytes, but it might also be an effect completely unrelated, acting in parallel of astrocytes.

I realize this is asking the authors to potentially acquire a new mice line, which might be difficult for a number of reasons. Therefore, if this mice line cannot be used by the authors, maybe they could take advantage of another experimental strategy, by for instance loading the astrocyte network surrounding the patched PN neurons with BAPTA, to buffer the response of astrocytes to both SST and GABA? This would suffice in supporting the authors main conclusions about the importance of astrocytes in regulating SST interneurons-PN synapses through purinergic signaling following astrocytes activation by GABA and SST. There are several studies using such approaches, including some cited in the manuscript (e.g. Serrano et al., 2006).

MINOR CONCERNS:

General comments:

  1. I think the authors could display individual cells values wherever possible, as this is now becoming a standard in the field. This would allow for a better appreciation on the cell to cell variability without interfering with the authors conclusions.
  2. I think the authors should work on the main text to make it more concise and avoid the use of lengthy sentences which can be hard to follow, requiring the reader to do multiple pass before understanding the authors points. I think that each sentence should convey just one idea, wherever possible.
  3. There are a few spelling mistakes in the manuscript to correct (e.g. line 317, t missing in light).
  4. I think the authors should provide proof their AAV strategy indeed lead to an SST interneuron- or astrocyte-specific expression of ChR2-mCherry or GCaMP6f, respectively. An immunohistochemistry experiment with astrocytes and SST interneurons markers coupled with mCherry or GFP antibodies and cell counting for colocalization of the respective transgenes with cell type markers would answer that concern. I understand other studies have used similar AAVs, but I think the authors cannot exclude the possibility of batch effects using different AAV solutions which can lead to a lack of cell type specificity. The material and method section explain that in a set of experiments, astrocyte staining with SR101 was done to confirm astrocyte specific expression of GCaMP6f. The authors should display these data, in Figure 2 for instance.

Abstract:

  1. In the abstract, the sentence ‘the role of astrocytes at GABAergic synapses have been investigated’ is a bit a stand alone statement. We don’t know if this refers to past work or the work of the author presented in the manuscript.

Results:

  1. Figure 1C: The legend says PN were patched, but the scheme shows a SST interneurons being patched. Please correct.
  2. Figure 2: The authors could increase the size of the ΔF/F0 traces.
  3. Figure 3-5, 10: I am not satisfied with the way statistics are displayed. We have no idea if the symbols displayed above the barplots represent the comparison to baseline 1 (bsl1) or baseline 2 (bsl2), or an average of both? Similarly, was a p-value correction for multiple comparisons used when comparing S1, R1, S2, R2 to what I guess is Bsl1? This is important and should not be neglected. This type of precisions can simply be written in the figure legends.
  4. Similarly, I am wondering if it could be interesting statistically compared similar stimulations across different experimental groups, for instance: control S1 (Figure 3) versus S1 + CGP 55845 (Figure 4) ; control S1 versus S1 + CYN 15406 (Figure 5), and similarly for baselines, R1, S2, R2. This could reveal interesting information on already available data. Another way to say this, is that the author could use statistics to back up their affirmations, for instance when saying, line 294-298: ‘in contrast to the sustained response observed in the absence of CYN15406, the astrocyte Ca2+ signals returned to basal levels during the recovery time in the presence of CYN 15406 (Figure 5)’. By directly comparing the values of R1 across different antagonist treatment groups using a statistical test, the authors could strengthen their observations, by having both within and between groups comparisons. The authors could indicate such statistics directly in the text, without having to change all figures.
  5. Figure 4: The significant decrease of events frequency following S1 (so during R1) under CGP 55845 for microdomains is puzzling me. Can the authors provide an explanation for this observation?
  6. Figure 5B: The authors do not comment on the fact that only in this pharmacological condition of SSTR blockade is the amplitude of calcium in microdomains increased following S1, S2 and R2, while this is not the case in the control condition (Figure 3). This is also puzzling me, can the authors provide an explanation for this observation?
  7. Figure 6-9: Similarly to my minor concern number 8, I think the authors should at the minimum state in the legend that the statistical comparisons displayed directly above the barplots are done against the average of the mean IPSC amplitude in basal conditions. The authors might also solve the problem by displaying the baseline average in the barplots, which would also allow to display the baseline variability. I would also be in favor of grouping all results from Figure 6-9 in one main figure, because as it stands, the data of Figure 6 are actually displayed four times in the paper, which leads to confusions as to whether the experiment in control condition was done once or four times. Looking at the data, it seems they were done only once, but displaying them only once would clarify this. Grouping this figures would also render easier for the reader to see at a glance that SSTR signaling has an action opposite to GABAb / A1R signaling on IPSC amplitude.
  8. Figure 6: The authors might want to show that the ‘low’ stimulus train (termed basal) is not evoking any response in astrocytes, compared to the intense train used in figure displaying astrocytes calcium imaging data.
  9. Line 339-340: ‘This suggests that in the absence of astrocytes recruitment, the reduction of inhibition was prevented…’: I think the authors meant that in the absence of astrocytes recruitment through GABAb signaling, the reduction of inhibition was strengthened, not prevented. The authors might also chose to call this phenomenon disinhibition in all cases, as mixing ‘reduction of inhibition’ and ‘disinhibition’ in the same sentence creates unnecessary complexity in the understanding of the text.
  10. Line 384: ‘…activation of postsynaptic A1R favors and increase in IPSCs that couterbalances…’: The authors should precise an increase in ‘IPSC amplitude’. There are similar cases in the rest of the manuscript.
  11. Line 386-387: Why are the authors arguing that SST has pre-synaptic effect? Their summary scheme (Figure 11) is showing presence of the SSTR on both pre- and post-synaptic elements, which is more correct in my opinion.
  12. Figure 10: The authors should be careful in saying that A1R blocker has no effect on the calcium response of astrocytes to SST interneuron activation. For instance S1 has an effect on the amplitude of calcium transients in proximal processes in control condition, which is absent when A1R are blocked. R2 effect on microdomains event frequency is also reduced when A1R are blocked, based on the authors statistics. I still agree with the main conclusion that A1R are not involved in astrocytes responses to SST interneurons activation, but the careful analysis of calcium transients performed by the authors, which I commend, reveals subtle differences with the control conditions that cannot just be discarded.
  13. Line 380-390: The data indeed favor an action of adenosine downstream of astrocytes, but I think the authors cannot really argue these effects are pre- or post-synaptic, and the same goes for the site of action of SST on SSTR. I am generally a bit disturbed by the affirmations of observed effects being pre- or post-synaptic In the ‘results’ section without experimental proofs, especially given that the authors are more careful on this matter in the discussion and call for more studies to localize the position of SSTR and A1R (Lines 492-497 / Lines 517-519). The results should be free of speculations, which should only be present in the discussion, in my opinion.

As far as my understanding goes, with the authors’ experimental model of evoked IPSC, a change in the IPSC amplitude could be due to either decreased GABA release (pre-synaptic) but also a decrease in the number of GABAa at the post-synaptic site or a decreased conductance of GABAa ionotropic receptors. I think that without clear experimental evidence, it is hard to define exactly where at the synapse are SST and adenosine acting. Maybe such evidence is already in the data or cited literature and I failed to see it, but then the authors should more clearly justify their affirmations on these matters.

Discussion:

  1. I think the authors should discuss whether their model of intense firing of SST interneurons is physiologically relevant, are there data in the literature showing similar firing frequency in vivo?
  2. Line 474-476: ‘Because neuronal GABABRs mediate inhibitory effects, we rule out the possibility that the stronger disinhibition observed in the presence of the GABABR antagonist with respect to controls, could be due to activation of neuronal GABABRs’ I have a hard time understanding the authors logic. Since indeed, GABAb mediates inhibitory effects in neurons, blocking their activation would lead to less inhibitory tone in neurons, which is exactly what the authors found in Figure 7. I apologize if I missed something here. I think this is the type of affirmation that would be possible if authors found similar effects of GABAb antagonist and astrocyte-specific GABAb KO on IPSC amplitude.
  3. Line 523-526: Linked to a previous comment, I am not convinced the data of the authors support the idea that SST release act on pre-synaptic SSTR to decrease the release of GABA. Are there results from the authors or from the literature showing SSTR in VCx are strictly pre-synaptic? This would mean that this is a form of autocrine/paracrine signaling of SST interneurons to SST interneurons? This would be interesting if this is the case, but existing data in support of such hypothesis should then be better discussed. Also, the scheme in Figure 11 is in agreement with my argument, as it shows both pre and post-synaptic localization of SSTR.

  1. Globally, I think the discussion could also briefly address the issue of how their results integrate in the larger context of heterosynaptic modulation by astrocytes: How can the activation of astrocytes by activity at SST interneurons GABAergic synapses, such as the one studied here, influence another, unrelated synapse in the grasp of the same astrocyte?
  2. Figure 11: This scheme is a bit too simplistic, with the SST interneuron making what looks like a synapse onto its own synapse. Maybe the SST neuron cell body could simply be removed, as its is not bringing any crucial information. Why is the SSTR absent from the astrocyte? It is presented on the astrocytes in Figure 5, so why not in Figure 11?

References:

  1. Some references contain the DOI, some others don’t, this should be made consistent according to the journal standards.

Reviewer 3 Report

This study reveals that SST GABAergic neuron-glia interaction regulates inhibitory synaptic transmission in the visual cortex. Although their findings are mainly based on the previous study from the corresponding author's lab (Mariotti et al., 2018, Nature Communications), this study has additionally provided consequences of astrocytic Ca2+ elevation by SST GABAergic neurons: ATP/adenosine-mediated elevation of inhibitory tone in nearby pyramidal neurons. The authors concluded that the GABA and SST release from SST neurons evokes both neuronal and astrocytic mechanisms, but the effects of such mechanisms on inhibitory synaptic transmission are opposite. Because this study is well designed and performed, their conclusion seems to be valid. They extensively discussed functional significances of this SST GABA neuron-astrocyte-pyramidal neuron relationship (possible involvements in sensory processing and perception) and molecular mechanisms that might be implicated in postsynaptic A1R signaling affecting inhibitory tones in the pyramidal neurons. 

I do not feel either additional experiment or substantial revision is required for publication in the Journal but only suggest authors review their manuscript and correct their minor typos (ex. 1um -> 1uM in line 356) before publication.

Reviewer 4 Report

Henriques VJ et al. used 2-photon Ca2+ imaging experiments and patch-clamp recording techniques with several antagonist to explore the signaling between Somatostatin (SST)-releasing GABAergic interneurons and astrocytes in mouse vCortex. They found that astrocytes modulate Sst interneuron signaling in vCtx. It is interesting. I have several concerns about this manuscript.

  1. Authors injected AAV viruses at P1-P2 and collected slices after 2 weeks as your method and result showed, but your Figure1A legend show 2-3 weeks after injection. I feel a little confuse about this. And usually AAV virus injection need to wait at least 3 weeks even longer. Of course, your image show it is expressed.

  1. One thing very confused to me that your Figure1C legend shows “whole-cell patch-clamp recordings from layer II/III PNs “. but on your figure1B and1C shows recording from Layer V Sst interneuron. I do not fully understand why you only record layer II/III PNs, I realize that Layer V PNs easier to tell and bigger.

  1. Authors used GABAB antagonist CGP 55845 at pretty high concentration. Usually people use 1 um. Your concentration is 5 time more. Is it possible to cause cell not very healthy? Try to lower concentration to see what happens.

  1. Authors used SSTRs antagonist CYN 154806, which mainly inhibits SSTR2. Did you try SSTR1 antagonist SRA 880? And your CYN 154806 concentration is also pretty high.

  1. Based on Rudy B et al 2011 review. Most of Sst interneuron distribute on layer V & VI. As your figure1B right upper panel show more like Layer V&VI. but your Figure 1C show as layer IV/V.

  1. As Su et. 2020 J. Neurosci. paper showed Sst has 2 types. Type I Sst mainly on infragranular (V & VI) but type II on supragranular (II & III). I am curious if astrocytes specifically modulate Type I Sst or regulate all type of Sst interneuron.

  1. Authors confirmed that astrocyte response to SST interneuron stimulation 274 is mediated by GABABR activation. I am curious that whether other GABA receptor is involved. Did you try it?